# Effect of Creatine Monohydrate Supplementation on Macro- and Microvascular Endothelial Function in Older Adults: A Pilot Study

**DOI:** 10.3390/nu17010058

**Published:** 2024-12-27

**Authors:** Holly E. Clarke, Neda S. Akhavan, Taylor A. Behl, Michael J. Ormsbee, Robert C. Hickner

**Affiliations:** 1Department of Health, Nutrition, and Food Sciences, Florida State University, 120 Convocation Way, Tallahassee, FL 32306, USA; holly.clarke.1992@gmail.com (H.E.C.); mormsbee@fsu.edu (M.J.O.); 2Department of Kinesiology & Nutrition Sciences, School of Integrated Health Sciences, University of Nevada, Las Vegas, NV 89154, USA; neda.akhavan@unlv.edu; 3Department of Applied Management, Flagler College, St. Augustine, FL 32084, USA; tbehl@flagler.edu; 4Department of Biokenetics, Exercise and Leisure Sciences, School of Health Science, University of KwaZulu-Natal, Durban 4041, South Africa; 5Institute of Sports Sciences and Medicine, Florida State University, Tallahassee, FL 32306, USA

**Keywords:** nutraceutical, creatine monohydrate, vascular health, vascular aging, endothelial function, oxidative stress

## Abstract

**Background/Objectives:** A pilot study was conducted to investigate the effect of four weeks of creatine monohydrate (CrM) on vascular endothelial function in older adults. **Methods:** In a double-blind, randomized crossover trial, twelve sedentary, healthy older adults were allocated to either the CrM or placebo (PL) group for four weeks, at a dose of 4 × 5 g/day for 5 days, followed by 1 × 5 g/day for 23 days. Macrovascular function (flow-mediated dilation [FMD%], normalized FMD%, brachial-ankle pulse wave velocity [baPWV], pulse wave analysis [PWA]), microvascular function (microvascular reperfusion rate [% StO_2_/sec]), and biomarkers of vascular function (tetrahydrobiopterin [BH_4_], malondialdehyde [MDA], oxidized low-density lipoprotein [oxLDL], glucose, lipids) were assessed pre- and post-supplementation with a four-week washout period. **Results:** CrM significantly increased FMD% (pre-CrM, 7.68 ± 2.25%; post-CrM, 8.9 ± 1.99%; *p* < 0.005), and normalized FMD% (pre-CrM, 2.57 × 10^−4^ ± 1.03 × 10^−4^%/AUC_SR_; post-CrM, 3.42 × 10^−4^ ± 1.69 × 10^−4^%/AUC_SR_; *p* < 0.05), compared to PL. Microvascular reperfusion rates increased following CrM (pre-CrM, 2.29 ± 1.42%/sec; post-CrM, 3.71 ± 1.44%/sec; *p* < 0.05), with no change following PL. A significant reduction in fasting glucose (pre-CrM, 103.64 ± 6.28; post-CrM, 99 ± 4.9 mg/dL; *p* < 0.05) and triglycerides (pre-CrM, 99.82 ± 35.35; post-CrM, 83.82 ± 37.65 mg/dL; *p* < 0.05) was observed following CrM. No significant differences were observed for any other outcome. **Conclusions:** These pilot data indicate that four weeks of CrM supplementation resulted in favorable effects on several indices of vascular function in older adults.

## 1. Introduction

Cardiovascular disease (CVD) remains the leading cause of mortality, morbidity, and increased healthcare costs in the United States [1]. While CVDs are etiologically complex, age remains the primary non-modifiable risk factor and predictor of CVD risk across the lifespan [2]. One common pathological process shared by CVDs and age-related diseases is a decline in vascular health, commonly known as vascular aging, characterized by arterial wall thickening, loss of distensibility, accumulation of oxidative stress, and a phenotypic shift towards vascular endothelial dysfunction. Endothelial dysfunction manifests in a pro-oxidative and pro-inflammatory environment with decreased nitric oxide (NO) bioavailability, increased oxidized low-density lipoprotein (oxLDL), and reduced vasomotor function, all of which elevate cardiovascular risk [3]. During vascular aging, these effects are compounded by the upregulation of adhesion molecules, elevated pro-inflammatory cytokines, and increased oxidative stress from reactive oxygen species [3]. Aging also impairs endothelial production of endothelial nitric oxide synthase (eNOS) enzymes, crucial for NO production, and can lead to alterations in extracellular matrix components such as collagen and elastin, resulting in increased vascular stiffness [3]. Collectively, these molecular changes significantly heighten vascular rigidity and dysfunction, exacerbating cardiovascular disease risk, especially among older adults. While the impact of vascular aging on large vessels within the body is commonly discussed, vascular aging similarly affects the microvasculature. Deteriorations in the microvasculature have also been linked to the advancement of CVDs [4,5].

While pharmaceuticals are often utilized to mitigate CVD risk, they are not without their limitations. Encumbering side effects, financial costs, and issues of medical adherence and accessibility are all limitations of pharmaceutical treatments. Thus, there is a need for safe and effective strategies for the prevention of CVDs. The incorporation of dietary supplements rich in antioxidant and anti-inflammatory properties has been shown to help attenuate CVD risk [6].

Creatine (Cr) is an amino acid compound, synthesized endogenously and consumed exogenously through the ingestion of red meat and seafood. The endogenous synthesis of Cr (~1 g/day) requires three amino acids: glycine, arginine, and methionine, together with two primary enzymes: _L_-arginine:glycine amidinotransferase and N-guanidinoacetate methyltransferase. Intracellularly, Cr can exist either in its free form or its phosphorylated form, creatine phosphate (PCr), via its enzymatic interaction with creatine kinase (CK). Following years of pioneering research, it is clear that Cr, along with PCr and CK, plays a crucial role in energy metabolism [7]. Cr and PCr contribute to rapid adenosine triphosphate (ATP) energy provision during periods of high ATP demand, ATP replenishment, and the shuttling of high-energy phosphates from areas of production to sites of utilization. Hence, the Cr/PCr system functions as a temporal high-energy phosphate buffer and spatial high-energy phosphate shuttle [8].

While supplementation with Cr has been repeatedly shown to confer physical performance benefits, evidence suggests that Cr may also serve a therapeutic role in various diseases, such as myopathies [9], neurodegenerative disorders [10], and metabolic dysfunctions, such as diabetes [11,12]. Cr has also been shown to exhibit non-energy-related properties, such as stabilizing mitochondrial efficiency [13] and serving as a potential antioxidant [14,15], anti-inflammatory agent [16], and lipid-lowering agent [17]. Moreover, Cr, PCr, CK isoenzymes, and Cr transporters are all expressed within endothelial cells that line the vascular tree, with the intricate Cr/PCr system playing a documented role in the delivery of high-energy phosphates to ion channels, which in turn contribute to the maintenance of vascular tone [18]. Despite these novel clinical properties and the presence of Cr within vascular-specific cells, there is very little in-human data pertaining to the impact of Cr on vascular health in older populations or those at risk of CVD.

The primary purpose of this pilot study was to investigate the effect of four weeks of Cr supplementation on the function of the macro- and microvasculature in sedentary, healthy older adults. We hypothesized that Cr supplementation would result in an improvement in both macro- and microvascular function, in part through a decrease in oxidative stress.

## 2. Methods

### 2.1. Study Participants

Men and postmenopausal women, aged 50–64 years, reporting low habitual physical activity levels (≤150 min per week of moderate activity), were recruited for this pilot study. Older adults with low habitual physical activity levels were selected as the population of interest because lifelong exercisers exhibit greater arterial compliance and sustained vascular health than sedentary older adults [19]. Exclusion criteria included preexisting kidney abnormalities, uncontrolled hypertension (>140 mmHg/>90 mmHg), the presence of CVD, metabolic disease, or other chronic diseases; taking medications that would interfere with central or peripheral circulation; current consumption of Cr; and consumption of any form of tobacco product. During the on-site screening, in addition to a medical history questionnaire, a fasted venous blood draw was collected to determine kidney health using a Piccolo Xpress Chemistry Analyzer (Abaxis Inc., Union City, CA, USA). Kidney health was determined by assessing total blood creatinine, estimated glomerular filtration rate (eGFR), and blood urea nitrogen (BUN). Participants were deemed eligible if creatinine was between 0.6–1.2 mg/dL, if eGFR was >60 mL/min, and if BUN was between 6–24 mg/dL.

### 2.2. Study Design

In this pilot study, a randomized, double-blind, placebo-controlled, crossover design was utilized. Eligible participants were randomly allocated in a 1:1 ratio via a computerized randomization generator to one of two order sequences: creatine monohydrate (CrM) to placebo (PL) or PL to CrM. All those involved in the study were blinded to the identity of each supplement. The identity of each supplement was kept confidential by an individual independent of study involvement until final statistical analyses were performed. All variables were measured pre- and post-CrM and PL. Prior to crossover, participants completed a four-week washout period. Both CrM and PL supplementation were consumed following the same dosing protocol: 4 × 5 g/day for 5 days, followed by 1 × 5 g/day for 23 days. Participants were asked to attend all visits having abstained from food, caffeine, exercise, and alcohol for ≥12 h prior to the scheduled start time. Additionally, participants completed an International Physical Activity Questionnaire (IPAQ-L) and a 3-day Food Log (analyzed using Food Processor version 11.1) prior to each testing visit to ensure habitual consistency throughout their study involvement. This study was approved by the Institutional Review Board of Florida State University (code: STUDY00000764, date: 12 April 2019). Participants were informed of the risks, requirements, and primary purpose of the study before a written informed consent was obtained. This study complied with all ethical principles for research involving human subjects (ClinicalTrials.gov Identifier: NCT05014659).

### 2.3. Supplementation Protocol

Both CrM and PL (maltodextrin) were sourced from Dymatize Nutrition (Dymatize^®^ Nutrition, Dallas, TX, USA). The monohydrated form of Cr was chosen over other existing forms (i.e., citrate, ethyl ester), as CrM is the only source of Cr that has substantial evidence supporting its superior bioavailability, efficacy, and safety [20]. Supplements were unflavored, with a similar appearance and texture, and were given to participants in powdered form, in sealed, blinded packaging. Participants were instructed to consume each supplement for four weeks (28 days), following a loading and maintenance protocol: 4 × 5 g/day for 5 days, followed by 1 × 5 g/day for 23 days. Supplementation with CrM following this loading protocol has been shown to significantly increase both plasma Cr concentrations and total Cr content of major muscles by 20–40% [20,21,22], while the subsequent maintenance dose of 5 g/day has been shown to sufficiently account for the day-to-day turnover of approximately 2 g Cr [20,23]. Furthermore, of importance, this dose of CrM has been shown to be safe and feasible in an older adult population [24]. Participants were instructed to dissolve and consume each dose in 8–16 oz. of water. A four-week washout period was utilized to separate supplement regimens. While there is great variation in Cr clearance between individuals, impacted by sex, body mass, renal function, and turnover [25,26], this washout period has been shown to sufficiently allow for a return to baseline muscle Cr stores in sedentary adults [21,27]. Compliance with supplement consumption was assessed with logs. The onset or presence of side effects was monitored using self-report forms.

### 2.4. Hemodynamic and Fluid Dynamic Assessments

Upon arrival for each visit, body weight (kg) and height (cm) were measured using a digital scale (Seca Corporation, Chino, CA, USA) and a wall-mounted stadiometer, respectively. Body mass index (BMI) was then derived using the formula: BMI = weight (kg)/(height [m])^2^. Participants were instructed to lie supine for the assessment of total body water (TBW [L]), intracellular fluid (ICF [L]), and extracellular fluid (ECF [L]) using an ImpediMed SFB7 bioimpedance spectroscopy device (ImpediMed Inc., Carlsbad, CA, USA). Following 15 min of supine rest, hemodynamic variables were measured using an automated blood pressure cuff (Omron Healthcare, Vernon Hills, IL, USA). Resting heart rate (HR), systolic blood pressure (SBP), diastolic blood pressure (DBP), and mean arterial pressure (MAP) were recorded.

### 2.5. Standardization of Vascular Assessments

Participants were asked to abide by standardized testing requirements prior to their visit: abstaining from food, alcohol, caffeine, and exercise for ≥12 h prior. Participants were also required to disclose any medication or vitamin use up to 72 h prior to the visit. All assessments were conducted by the same operator. Testing visits were completed at the same time of day (±1 h) to control for circadian rhythm.

### 2.6. Brachial Artery Flow-Mediated Dilation

Brachial artery flow-mediated dilation (baFMD) was assessed following published guidelines at the time of the study [28] and analyzed using commercially available software (Vascular Research Tools 6, Medical Imaging Applications LLC, Coralville, IA, USA). Prior to the assessment, resting heart rate and blood pressures were measured in duplicate, and participants were considered rested if the recorded heart rates and blood pressures were within steady-state limits of ±5 bpm and ±5 mmHg, respectively.

Brachial artery flow-mediated dilation and simultaneous mean blood velocity (MBV) were measured in duplex mode with a high-resolution doppler ultrasound system and a 12 MHz linear array transducer (Philips HD11XE, Philips Ultrasound, Bothwell, WA, USA). To assess FMD, a rapidly inflating blood pressure cuff (Hokanson E20, Hokanson, Bellevue, WA, USA) was placed on the upper forearm (~2 cm distal to the antecubital fossa) and inflated to 250 mmHg to occlude forearm blood flow. Ultrasound images and video were captured in real-time using commercially available video software (VIDBOX Inc., Austin, TX, USA) and saved for future analysis. Images and videos were captured at baseline, during occlusion, and post-occlusion (cuff release).

All videos were analyzed using edge-detection and wall-tracking software (Vascular Research Tools 6, Medical Imaging Applications LLC, Coralville, IA, USA). Shear rate (SR) was calculated as 8 × MBV/vessel diameter, and the SR area under the curve (AUC_SR_) from cuff release to peak diameter was quantified as an index of the stimulus for FMD [29]. FMD values are presented as relative (%) change from baseline to peak diameter and corrected for shear rate (FMD%/AUC_SR_) to account for inter-individual variability in shear stress.

### 2.7. Pulse Wave Analysis and Velocity Assessments

Pulse wave analysis (PWA) was measured using the AtCor SphygmoCor System (AtCor Medical, West Ryde, NSW, Australia). Participants were laid supine, and a high-fidelity applanation tonometer was positioned on the left wrist to detect the radial pulse. Ten seconds of high-quality arterial waveforms were captured, and a general transfer function was applied to generate central aortic pressure waveforms. Central pressure waveforms were characterized using several variables, including aortic systolic and diastolic pressure, augmentation pressure (AG), and central pulse pressure (cPP). The augmentation index (AIx), an indirect measure of arterial stiffness expressed as a percentage, was determined as: (AG/cPP) × 100. Due to the impact of heart rate on AIx, values were corrected for a standard heart rate of 75 bpm (AIx@75bpm). Pulse wave velocity (PWV) was measured using an Omron VP-1000 (Omron Healthcare, Vernon Hills, IL, USA). While supine, four pressure cuffs were placed around both upper arms and ankles. Electrodes were placed on both wrists to establish a baseline electrocardiogram, and a phonocardiogram was placed above the apex of the heart to detect heart sound signals. Following the automatic inflation of cuffs, simultaneous brachial and tibial artery pulse waves were recorded, and an average pulse wave transit time between sites was calculated. PWV was calculated as: distance (m)/pulse wave transit time (sec). Both left and right PWV were measured and reported.

### 2.8. Near-Infrared Spectroscopy Microvascular Assessment

Microvascular function was assessed using near-infrared spectroscopy (NIRS), with data collected concurrently during FMD. Prior to FMD cuff inflation, a portable NIRS device (Muscle Oxygen Monitor, Fortiori Design LLC, Hutchinson, MN, USA) was secured on the observed flexor carpi radialis muscle, located midway between the occlusion cuff and wrist. To reduce the impact of surrounding light, black self-adhesive bandage was used to secure the NIRS unit. NIRS measurements were collected in real-time continuously for the duration of the FMD procedure. Using muscle oxygen monitor software (version 1.3), raw data was then extracted and saved for future analysis. Time (seconds), average muscle tissue oxygen saturation (StO_2_), and real-time StO_2_ were utilized for data analysis. Baseline StO_2_ was determined using the average of the StO_2_ readings 30 s prior to cuff inflation. To assess microvascular reactivity in response to post-occlusion hyperemia, the reperfusion rate was determined as the upslope of the saturation signal (StO_2_%/sec) over the 10-s window immediately following cuff deflation. This 10-s reperfusion slope has been shown to be indicative of microvascular reactivity and recruitment and correlates with baFMD [30,31].

### 2.9. Blood Collection and Analyses

Fasting venous blood samples were drawn from the antecubital vein at each experimental visit. Blood samples (~6 mL each) were collected into green top plasma (lithium heparin), lavender top plasma (EDTA), and red top serum (no additive) vacutainers. Samples were then centrifuged, aliquoted into labeled microtubes, and stored at −80 °C for future analysis. Concentrations of malondialdehyde (MDA), oxLDL, and tetrahydrobiopterin (BH_4_) were measured in duplicate using commercially available enzyme-linked immunosorbent assays, according to the manufacturer’s instructions. MDA is a byproduct of lipid peroxidation, which occurs when reactive oxygen species (ROS) attack polyunsaturated fatty acids in cell membranes. The presence of MDA indicates oxidative damage to lipids, which can affect cell integrity and function. oxLDL is a low-density lipoprotein that has been oxidized through interactions with ROS, leading to modifications in lipid and protein components. This oxidative modification can occur within the arterial wall and increase the risk of atherosclerosis. Both increased MDA and oxLDL are associated with cardiovascular risk and thus were utilized as biomarkers of interest in this study. BH_4_ is a critical cofactor for eNOS and plays a vital role in maintaining vascular health through the production of NO, reducing oxidative stress, and protecting endothelial function; thus, it was chosen as a novel biomarker of interest. In addition, fasting blood glucose and lipids, including cholesterol, high-density lipoprotein, triglycerides (TGs), non-HDL cholesterol, low-density lipoprotein (LDL), and very LDL (vLDL), were also assessed using the Piccolo Xpress Chemistry Analyzer—Lipid Panel Plus disks (Abaxis Inc., Union City, CA, USA).

### 2.10. Statistical Analysis

The primary endpoint of this pilot study was FMD%. To determine a sufficient sample size for this primary endpoint, a power analysis was conducted using G*Power, F-test, ANOVA: repeated measures, within factors, with an alpha (α) of 0.05 and power (1—β) of 0.8. Using results published by Jones et al. [32], chosen for their comparable study design, population, and endpoints, a sample size of 12 participants was deemed necessary to detect a significant difference in FMD%. Given that this was a pilot study, no further power analyses were conducted for secondary endpoints.

Data were analyzed with the SPSS software package, version 25 (SPSS Inc., Chicago, IL, USA). The normality of the data was examined using the Shapiro-Wilk test. To identify potential outliers, studentized residuals were generated for each dataset and examined. To determine the success of the randomization of participants into either sequence group, baseline group characteristics were analyzed using independent *t*-tests. A 2 × 2 repeated measures analysis of variance (ANOVA) was utilized as the primary statistical method for all dependent variables: (time [pre- and post-] × group [CrM and PL]). This allowed for the identification of any group, time, or group × time interaction. If significant interactions were observed, subsequent Bonferroni post hoc analyses were performed. To determine whether a four-week washout period was sufficient, baseline values for dependent variables were compared using paired sample *t*-tests. Absolute change pre- to post-, or “delta” change (Δ) in scores, of all dependent variables was analyzed with paired *t*-tests to determine any statistical difference between groups (CrM vs. PL). All data are presented as mean ± SD, with a *p* < 0.05 indicating statistical significance. Effect size is reported for the primary endpoint of FMD% only.

## 3. Results

### 3.1. Participant Characteristics

Twelve eligible participants (6 men and 6 women) were fully enrolled and completed this crossover pilot study. Full participant recruitment and disposition are displayed in Figure 1. Descriptive participant characteristics are presented in Table 1. Participant characteristics were well balanced between randomized groups.

### 3.2. Hemodynamics and Fluid Dynamics

No statistical differences were observed between groups at pre-supplementation. Following CrM or PL supplementation, no changes were observed in HR, SBP, DBP, or MAP. Similarly, no change was observed in TBW, ECF, or ICF following either supplement.

### 3.3. Flow-Mediated Dilation

FMD and associated variables are presented in Table 2. Resting diameter was similar across all study visits and supplement groups, indicating consistent baseline measures. In addition, calculated AUC_SR_ were similar at all visits, indicating consistent vasodilatory stimuli.

Following PL supplementation, no significant change in FMD% was observed (pre-PL, 8.13 ± 2.76%; post-PL, 8.08 ± 2.07%); however, FMD% was significantly increased following CrM (pre-CrM, 7.68 ± 2.25%; post-CrM, 8.9 ± 1.99%; *p* < 0.005; Table 2 and Figure 2A). The effect size, as measured by Cohen’s d, was 0.4, indicating a medium effect of treatment. These observations persisted following the normalization of FMD% by shear rate. While no change was observed following PL (Pre-PL, 2.48 × 10^−4^ ± 9.28 × 10^−5^%/AUC_SR_; Post-PL, 2.38 × 10^−4^ ± 1.07 × 10^−4^%/AUC_SR_), supplementation with CrM resulted in a significant increase in normalized FMD (pre-CrM, 2.57 × 10^−4^ ± 1.03 × 10^−4^%/AUC_SR_; post-CrM, 3.42 × 10^−4^ ± 1.69 × 10^−4^%/AUC_SR_; *p* < 0.05; Figure 2B).

### 3.4. Pulse Wave Analysis and Velocity

There were no significant differences in any PWV or PWA variables between groups pre-supplementation, indicating sufficient washout. No significant changes in PWA or PWV were observed following CrM or PL supplementation.

### 3.5. Near-Infrared Spectroscopy

There was no significant difference observed between StO_2_ reperfusion rates pre-supplementation of CrM or PL, indicating sufficient washout and return to baseline between regimens. Following PL supplementation, no change in StO_2_ reperfusion was observed (pre-PL, 2.47 ± 1.4%/sec; post-PL, 2.11 ± 1.01%/sec); however, following CrM supplementation, a significant increase in StO_2_ reperfusion rate was observed (pre-CrM, 2.29 ± 1.42%/sec; post-CrM, 3.71 ± 1.44%/sec; *p* < 0.05; Figure 3).

### 3.6. Blood Analyses

No significant differences in plasma MDA, oxLDL, or BH_4_ between the CrM and PL conditions pre-supplementation were observed, indicating sufficient washout and return to baseline.

No significant changes were observed across all explored biomarkers. While a numerically greater reduction in MDA was observed following CrM supplementation versus PL (−79.21 ng/mL vs. 82.27 ng/mL, respectively), this was not statistically significant. Similarly, CrM supplementation resulted in a slightly larger numerical reduction in oxLDL versus PL (−0.27 μg/mL vs. −0.11 μg/mL, respectively); however, this was not significant. Finally, a greater numerical increase in BH_4_ concentration was observed following CrM compared to PL; however, this was not statistically significant (22.79 pg/mL vs. −25.94 pg/mL, respectively).

In regard to blood glucose and lipids, significant changes were observed. Following PL, there were no changes in fasting glucose (Pre-PL, 101.91 ± 7.53 mg/dL; post-PL, 102.82 ± 9.23 mg/dL); however, following CrM, a significant decrease in glucose was observed (pre-CrM, 103.64 ± 6.28; post-CrM, 99 ± 4.9 mg/dL; *p* < 0.05). Similarly, following PL, there were no changes in TGs (pre-PL, 91 ± 46.34; post-PL, 99.45 ± 45.17 mg/dL), whereas CrM supplementation led to a significant decrease (pre-CrM, 99.82 ± 35.35; post-CrM, 83.82 ± 37.65 mg/dL; *p* < 0.05). No significant changes were observed for any other analyzed lipid.

## 4. Discussion

In this pilot study, we investigated the impact of CrM supplementation on macro- and microvascular function in older adults. We observed that four weeks of CrM supplementation demonstrated favorable benefits in vascular function, as indicated by an increase in FMD% and normalized FMD. Additionally, CrM improved microvascular recruitment and reactivity, as indicated by an increase in StO_2_ reperfusion, and reduced circulating blood glucose and triglycerides. However, these improvements were not accompanied by changes in PWA, PWV, TBW, or cellular fluid distribution. Our hypothesis that four weeks of CrM consumption would reduce oxidative stress cannot be supported by the present data.

Following CrM supplementation, FMD% significantly increased by an absolute average of 1.22%. This improvement in endothelial function remained following normalization. Considering macrovascular dysfunction is strongly correlated with a higher risk of CVD, these preliminary findings are of interest. While FMD may not be a risk factor for CVD, per se, its association with endothelial function makes it a valuable prognostic tool [33]. In a large meta-analysis of 23 studies, across 14,753 patients, a significant inverse association between brachial FMD at baseline and CVD risk was observed [34]. Furthermore, it has been previously reported that for every 1% increase in FMD%, there is a resulting 13% decrease in the future risk of a CVD event [35]. Thus, considering the average improvement observed here, CrM supplementation may confer vascular-benefiting properties, thereby supporting vascular health. However, future research with larger sample sizes is warranted to confirm these findings. While the novelty of this study limits the ability to compare results with other studies, the magnitude of improvement in FMD% observed here is comparable to that seen following other nutraceuticals and pharmaceuticals, such as blueberries [36], coenzyme Q10 [37], and Amlodipine Besylate (antihypertensive) [38]. Therefore, future studies should look to concurrently compare CrM with other known vascular-benefiting supplements or therapeutics to directly assess whether the magnitude of improvement is comparable.

In regard to arterial stiffness, no changes were observed following CrM. This contrasts that reported recently by Aron et al., who observed an improvement in the cardio-ankle vascular index following Cr supplementation for 7 days versus PL among males aged 55–80 years [39]. Sanchez-Gonzalez et al. also reported significant reductions in PWV following 3 weeks of CrM supplementation; however, improvements were reported immediately following fatiguing bouts of isokinetic exercise [40]. No changes were observed at rest. In the current pilot study, we also assessed PWA. Following CrM supplementation, there were no changes in AG or AIx. Both AG and AIx are known to be influenced by, and correlate closely with, blood pressure and heart rate. In the current pilot study, no changes in resting SBP, DBP, or HR following CrM supplementation were observed. This lack of effect on hemodynamics in response to CrM has been similarly reported in other studies [39,41,42,43,44]. While this lack of influence on hemodynamics is not a negative outcome, it may explain the lack of impact of CrM on PWA variables.

In addition to an increase in baFMD, we observed novel, significant improvements in microvascular StO_2_ reperfusion following CrM, with an average increase of 1.42%/sec. These results corroborate those reported by other investigators. In an open-label study, Moraes et al. reported a significant improvement in capillary density and recruitment following 1 week of CrM supplementation, consuming 20 g/day [45]. Similarly, Van Bavel et al. also reported a significant improvement in microvascular density and reactivity following CrM supplementation at a dose of 5 g/day for 3 weeks [46]. Despite the variation in the length of CrM supplementation and assessment methodologies between the current pilot study and the previous studies, a consistent improvement in microvascular function was observed, thus supporting the potential vascular application of CrM.

Age-associated increases in oxidative stress have been shown to contribute to vascular pathologies. Due to the reported, albeit novel, antioxidant properties of Cr, we hypothesized that CrM supplementation may indirectly improve vascular health by reducing oxidative stress. This hypothesis was not supported here, as no significant changes in oxidative stress biomarkers were observed in this pilot study. In contrast to what was observed here, Rahimi et al. reported a significant MDA-lowering effect of CrM following acute exercise-induced muscle damage compared to placebo [47]. These findings were similar to those reported by Deminice et al. in an animal model of exercise-induced damage [17]. While Ahsan et al. reported a reduction in oxLDL-induced endothelial apoptosis following PCr supplementation in vivo [48], there is scarce literature regarding the impact of CrM on lipid peroxidation in clinical populations. Therefore, further investigation is required to explore the impact of CrM on oxLDL.

While no antioxidant effects were observed following CrM supplementation in the current pilot study, this may have been due to some methodological limitations of the current study. For example, only three oxidative stress biomarkers were investigated, whereas there are a multitude of markers with clinically validated assays available, that have shown a strong association with vascular health, such as C-reactive protein, inflammatory markers (IL-6), homocysteine, and adhesion molecules. Thus, the impact of CrM on other antioxidant enzymes or pathways warrants further investigation before any further inferences should be made. Furthermore, given this was a pilot study, the resulting sample size was relatively small and not powered sufficiently to detect significant changes in blood biomarkers. This further supports the need for future investigations with larger cohorts.

Elevated plasma glucose and TGs have been shown to contribute to increased CVD risk [49]. In the current study, both fasting glucose and TGs were significantly reduced following four weeks of CrM, with fasting glucose levels decreasing following CrM supplementation from prediabetic levels, per the American Heart Association (100–125 mg/dL), to a normal, healthy range (103.64 ± 6.28 to 99 ± 4.9 mg/dL), compared to no change with PL (101.91 ± 7.53 mg/dL to 102.82 ± 9.23 mg/dL). Current literature surrounding the impact of CrM supplementation alone on glucose regulation is varied, with some studies reporting a decrease in fasting glucose [50,51] and others reporting an increase or no change [45,46,52,53]. Proposed mechanisms in which CrM may contribute to glycemic improvements include CrM’s role in GLUT-4 expression and translocation^11^; however, further research is needed in older populations. We also report a significant reduction in TGs following CrM. These reductions are similar to that reported by Earnest et al., who found a significant reduction in triacylglycerols and vLDLs following 12 weeks of CrM supplementation in those with hypercholesterolemia [54]. Considering the strong correlation between hyperglycemia and hypertriglyceridemia and CVD risk, these observed reductions are of merit and further highlight the clinical potential of CrM supplementation.

While the mechanism(s) responsible for our observed vascular improvements cannot be fully concluded, there are several potential explanations. As mentioned previously, NO bioavailability is paramount for healthy endothelial function. NO, in the vasculature, is synthesized by eNOS, and requires the investment of _L_-arginine and oxygen. Bode-Böger et al. and Yin et al. found that _L_-arginine supplementation improved endothelial function, with the hypothesis that increased _L_-arginine bioavailability led to a greater production of NO [55,56]. The endogenous synthesis of Cr requires a considerable amount of _L_-arginine, with 20–30% of arginine’s amidino groups being utilized by this daily process. Therefore, supplementing with Cr may serve to spare _L_-arginine stores, allowing for greater NO synthesis. In addition to having the necessary substrates, the activation of eNOS is also vital. There are a variety of molecular signals that contribute to the activation of eNOS. One of the primary signaling pathways is the phosphatidylinositol 3-kinase/protein kinase B/eNOS (PI3K/Akt/eNOS) pathway. Ahsan et al. found, in vitro, that following PCr supplementation, there was a significant increase in PI3K/Akt/eNOS activity and NO production [48]. Given the part Cr may play in the synthesis of NO and the correlation between increased NO availability and FMD%, while speculative, the improvements observed in this current study to FMD may have been due to, in part, CrM supplementation.

Despite the role of NO in the macrovasculature, within the microvasculature other vasoactive mediators are believed to play a greater role. Cracowski and colleagues report that cutaneous microvascular flow is mediated not so much by NO, but by endothelial-derived hyperpolarization factors (EDHFs), a vascular phenomenon mediated by the intricate regulation of calcium-dependent and ATP-dependent potassium channels [57]. Although further investigation is needed, the energy-related benefits of Cr may contribute to the regulation of these vascular-specific ATP-dependent channels. As demonstrated in studies by Dzeja and Terzic and Selivanov et al., phosphotransfer enzymes such as creatine CKs are coupled to ion channels and pumps, including the ATP-sensitive potassium pump (KATP) [58,59]. This pump plays a crucial role in regulating potassium movement in endothelial cells and vascular smooth muscle cells (VSMCs). Given that the KATP pump is ATP-dependent, the ability of the Cr/PCr system to supply high-energy phosphates likely contributes to the intricate regulation and function of this pump. Therefore, although speculative, increased intracellular levels of Cr in tissues such as the endothelium—potentially through Cr supplementation—could help regulate KATP channels. This regulation could thereon assist in times of hyperemia to hyperpolarize neighboring VSMCs, enhance EDHF-mediated vasodilation, and ultimately contribute to improved vascular tone and health. These vascular-specific mechanisms, of which Cr may serve a valuable role, are just a few of the proposed pathways in which Cr may influence vascular function [60,61]. Given the improvements observed in this pilot study, future studies are warranted to further explore the vascular potential of Cr, with an emphasis on identifying primary mechanisms of action.

There are some limitations that should be taken into consideration when interpreting results. While the current study addresses a novel research question, this was an exploratory pilot study with a relatively small sample size. This small sample size may have increased the likelihood of selection bias, thus limiting whether the sample was representative of the broader population. This pilot was powered for our primary endpoint, FMD%. While we did achieve an appropriate sample size to detect a significant difference in FMD% between treatment arms, no further power analyses were conducted, thus limiting the ability to detect significant changes in other variables, which were likely underpowered. In addition, no covariate analyses were performed; however, with the lack of significant difference between treatment groups, we do not believe that findings would have been impacted. Furthermore, this pilot study lacked any assessment of Cr concentrations either in the plasma or muscle. In addition, while a washout period of four weeks is commonly used in Cr studies, due to the variability in absorption and turnover, a longer washout period of ≥6 weeks may be more appropriate in future trials in certain populations. Finally, while we used standardized FMD procedures that were standard practice at the time of study approval, FMD procedures and guidelines are continuously evolving. Thus, any future studies should ensure they use updated methodologies where applicable.

## 5. Conclusions

In summary, we present novel data demonstrating that four weeks of CrM supplementation led to favorable changes in both macro- and microvascular function in older adults, paralleled with reductions in blood glucose and TGs, suggesting the potential vascular benefits of CrM supplementation in this population. Considering the continued impact of CVD in the US, identifying accessible, affordable, and feasible strategies to benefit vascular health is crucial. Given the broad application of Cr, in addition to the potential vascular benefits seen here, Cr could be a beneficial supplement of choice to help mitigate age-related disease. Moreover, no adverse events or side effects following CrM supplementation were observed, nor was there any impact upon hemodynamics or body fluid distribution, thus supporting the safety of Cr in this population. Given our observations, with the caveat of being a pilot study, future trials are warranted to apply our findings. More comprehensive trials, with larger sample sizes that are more representative of the broader population, are needed to gain a deeper understanding regarding Cr’s role in vascular health. Future studies may also benefit from exploring increased intervention periods to determine the feasibility of the day-to-day incorporation of Cr supplementation. Furthermore, future research should look to identify the mechanism(s) by which CrM may be eliciting these vascular-benefiting properties.

## Figures and Tables

**Figure 1 nutrients-17-00058-f001:**
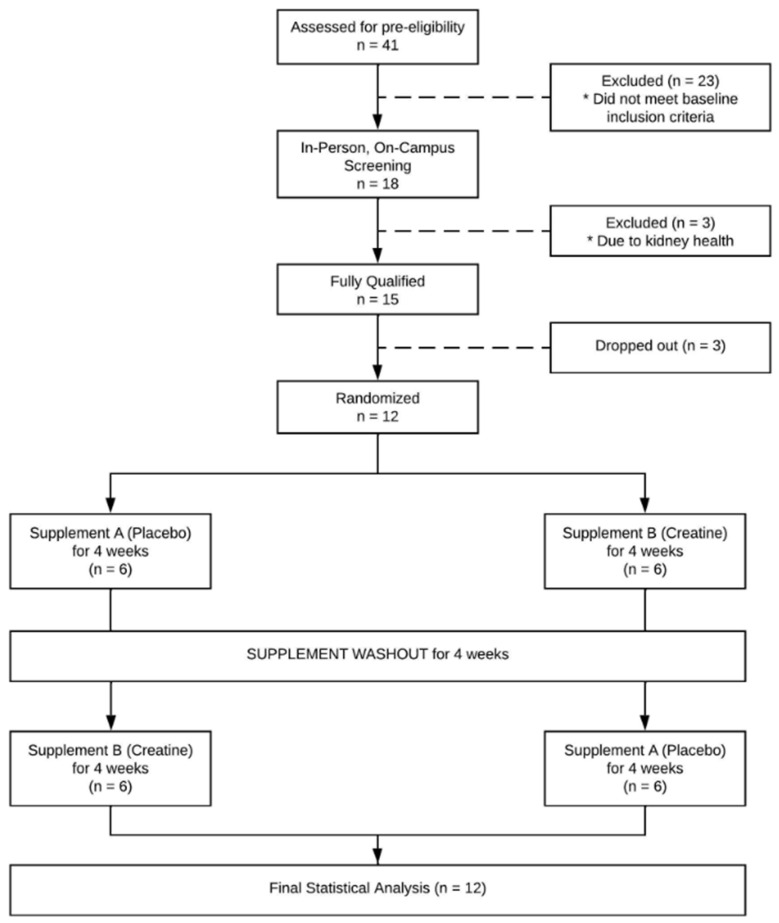
Participant flow through study. * denotes reason for exclusion.

**Figure 2 nutrients-17-00058-f002:**
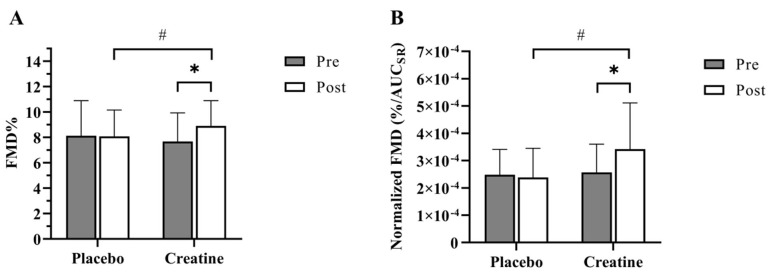
FMD% and normalized FMD% results. Data are expressed as mean ± SD. (**A**) Average flow-mediated dilation (%) pre- and post- each supplement. (**B**) Average normalized flow-mediated dilation (%/AUCSR) pre- and post- each supplement. * denotes a significant difference from the corresponding baseline (pre-), *p* < 0.05. # denotes a significant difference between post-treatment values, *p* < 0.05.

**Figure 3 nutrients-17-00058-f003:**
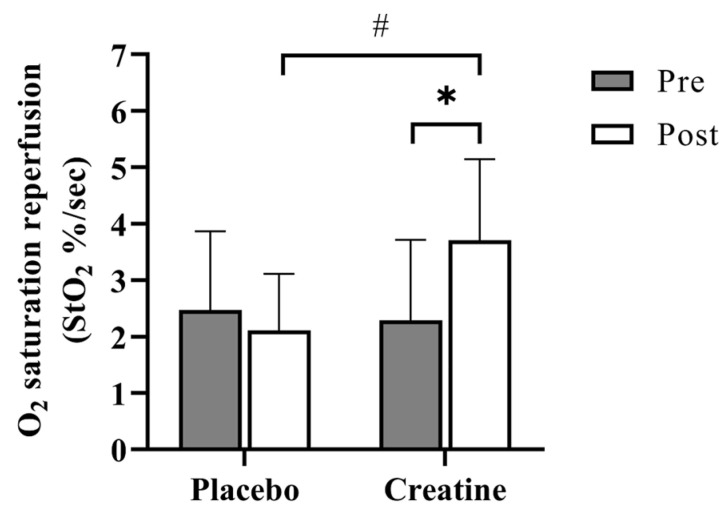
Average StO_2_ reperfusion results. Data are expressed as mean ± SD. * denotes a significant difference from the corresponding baseline (pre-), *p* < 0.05. # denotes a significant difference between post-treatment values, *p* < 0.05.

**Table 1 nutrients-17-00058-t001:** Participant descriptive characteristics.

		Randomized Group Sequence		*p* Value
	Overall	PL—CrM	CrM—PL	
N (Male/Female)	12 (6/6)	6 (4/2)	6 (2/4)	-
Age (years)	58.3 ± 3.4	59 ± 2.7	57.5 ± 4.1	0.473
Height (cm)	171.1 ± 9.0	171.7 ± 8.2	170.5 ± 10.4	0.824
Weight (kg)	75.0 ± 17.8	75.6 ± 17.9	74.5 ± 19.4	0.917
BMI (kg/m^2^)	25.6 ± 5.6	25.7 ± 5.9	25.5 ± 5.9	0.962

PL, placebo; CrM, creatine monohydrate; BMI, body mass index. All values are mean ± SD.

**Table 2 nutrients-17-00058-t002:** Changes in flow-mediated dilation and associated variables.

Variable		Placebo	Creatine
Resting Diameter (mm)			
	Pre	4.10 ± 0.73	4.21 ± 0.81
	Post	4.11 ± 0.8	4.18 ± 0.82
	Δ	0.01 ± 0.15	−0.13 ± 0.13
Absolute Change (mm) *			
	Pre	0.33 ± 0.11	0.33 ± 0.12
	Post	0.34 ± 0.11	0.38 ± 0.12 ^†‡^
	Δ	0.003 ± 0.06	0.05 ± 0.04 ^⁋^
Time to Peak (sec)			
	Pre	43.5 ± 7.73	38 ± 8.52
	Post	41.25 ± 11.87	41.5 ± 11.92
	Δ	−2.25 ± 14.65	3.5 ± 13.71
FMD% *			
	Pre	8.13 ± 2.76	7.68 ± 2.25
	Post	8.08 ± 2.07	8.9 ± 1.99 ^†‡^
	Δ	−0.05 ± 1.73	1.22 ± 0.87 ^⁋^
Shear Stress (AUC)			
	Pre	35,579.73 ± 12549.4	32,087.04 ± 8209.87
	Post	35,675.45 ± 9752.13	30,650.48 ± 10,244.53
	Δ	95 ± 5881.29	−1436.56 ± 4362.63
Normalized FMD (%/AUC_SR_) *			
	Pre	2.48 × 10^−4^ ± 9.28 × 10^−5^	2.57 × 10^−4^ ± 1.03 × 10^−4^
	Post	2.38 × 10^−4^ ± 1.07 × 10^−4^	3.42 × 10^−4^ ± 1.69 × 10^−4 †‡^
	Δ	−1.01 × 10^−5^ ± 4.60 × 10^−5^	8.44 × 10^−5^ ± 9.47 × 10^−5 ⁋^

Δ, delta change in scores; FMD%, flow-mediated dilation %; AUC, area under the curve; SR, shear rate; SD, standard deviation. * sig. interaction (group × time) for the dependent variable, *p* < 0.05; ^†^ sig. different from the corresponding baseline (pre), *p* < 0.05; ^‡^ sig. different from post-placebo, *p* < 0.05; ^⁋^ delta change sig. different from placebo delta change, *p* < 0.05; Data in mean ± SD.

## Data Availability

The data supporting the conclusions of this pilot study are presented in the current article. Additional raw data may be made available upon request to the authors, due to technical and timing limitations. Additional inquiries can be directed to the corresponding or lead author.

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
