# Peer review of "Effect of Creatine Monohydrate Supplementation on Macro- and Microvascular Endothelial Function in Older Adults: A Pilot Study"

_nutrients, 2024, doi:10.3390/nu17010058_

Round 1
Reviewer 1 Report
Comments and Suggestions for Authors
I have had the opportunity to review this pilot investigation examining the effects of creatine monohydrate supplementation on macro- and microvascular endothelial function in older adults. The manuscript presents an innovative exploration of therapeutic potential in vascular health management, demonstrating both methodological rigor and clinical relevance. While the pilot nature of the study appropriately tempers the breadth of its conclusions, the investigation provides valuable preliminary evidence warranting further systematic investigation.Abstract: The abstract demonstrates methodological rigor in its presentation of this pilot investigation. However, I would suggest enhancing the precision of certain statistical presentations. While the authors appropriately report confidence intervals for key findings, the presentation of effect sizes could be more systematically structured. The inclusion of specific p-values enhances scientific transparency, though standardization of statistical reporting formats would strengthen the abstract's academic rigor.
Introduction: The authors present a well-crafted theoretical framework, effectively establishing the clinical significance of vascular dysfunction in aging populations. The progression from broad cardiovascular implications to specific creatine mechanisms demonstrates sophisticated conceptual development. However, the literature review could be strengthened by:
- More explicit delineation of the mechanistic pathways linking creatine to vascular function
- Enhanced discussion of the temporal aspects of vascular adaptation
- More robust theoretical justification for the selected intervention duration
Methods: The methodological framework exhibits commendable attention to procedural detail and standardization. The crossover design strengthens internal validity, though several elements warrant consideration:
- The rationalization for the selected washout period could be more robustly supported with pharmacokinetic evidence
- The biomarker selection criteria could benefit from more explicit theoretical justification
- The statistical power calculations for secondary outcomes should be more clearly delineated
Results: The data presentation demonstrates appropriate statistical sophistication. The authors' approach to reporting both absolute and normalized FMD values enhances result interpretation. However:
- The presentation of subgroup analyses could be more systematically structured
- Effect size reporting could be more consistent across outcome measures
- The treatment of potential confounding variables could be more explicitly addressed
-
Discussion: The authors present a thoughtfully structured discussion that effectively contextualizes their findings within existing literature. However, several elements merit refinement:
- The mechanistic interpretations, while theoretically sound, could benefit from more explicit linkage to observed data patterns
- The clinical implications of the observed vascular improvements could be more comprehensively explored in relation to established cardiovascular risk thresholds
- The discussion of null findings (particularly regarding oxidative stress markers) warrants more nuanced consideration of methodological limitations versus true absence of effect
Limitations: The authors demonstrate appropriate scholarly restraint in acknowledging study limitations. However, I would suggest:
- More explicit discussion of potential selection bias implications
- Enhanced consideration of the temporal adequacy of the intervention period
- More detailed analysis of statistical power limitations for secondary outcomes
Conclusion: The conclusion appropriately reflects the pilot nature of the investigation while effectively highlighting its clinical implications. However, the authors could:
- More precisely articulate the hierarchical importance of their findings
- Provide more specific recommendations for future research design parameters
- Better delineate the clinical translation pathway for their findings
References: The bibliography demonstrates comprehensive coverage of relevant literature. However:
- Some more recent publications in vascular aging mechanisms could be incorporated
- The mechanistic literature regarding creatine's vascular effects could be expanded
- Additional methodological references regarding FMD assessment could strengthen the technical foundation
Author Response
Please see the document attached.

Reviewer 2 Report
Comments and Suggestions for Authors
The manuscript entitled ‘Effect of Creatine Monohydrate Supplementation on Macro- and Microvascular Endothelial Function in Older Adults: A Pilot Study’ written by Holly E. Clarke et al. presents interesting results regarding the effects of 4-weeks administration of creatine monohydrate in older people. Several macrovascular and microvascular parameters were evaluated. The authors have shown favorable effects of creatine monohydrate supplementation on some parameters, suggesting beneficial effects on vasculature in the studied population. Although the obtained results are interesting and have application potential, the number of studied subjects is limited and some details of the study design should be justified.
The manuscript is clear, scientifically sound, and presented in a well-structured manner; however, some improvements can be introduced (see comments below). The English language is correct and clear.
General concept comments
1. In the Abstract section, the change values are difficult to interpret without reference to means in groups. Therefore, I suggest the authors to, instead of showing delta values (∆), present differences between groups by showing both values obtained in the compared groups, e.g. FMD% 7.68 ± 2.25 vs. 8.9 ± 1.99, P < 0.05.
2. The justification for including participants with low habitual physical activity levels should be added. Furthermore, please explain the reason to choose monohydrate form from many forms of creatinine (pyruvate, citrate, malate, taurinate, phosphate, ethyl ester, magnesium creatine chelate). Providing this information will make the study design more clear.
3. The main limitation of the study is the low number of enrolled participants (n=12). I hope the authors continue the study with much larger cohorts.
I believe that my suggestions will help the authors improve the quality of their manuscript.
Author Response
Please see the document attached.

Round 2
Reviewer 1 Report
Comments and Suggestions for Authors
Thank you for the author response. The authors have brilliantly addressed all my observations and I have no further comments.